

# Be positive: customized reference databases and new, local barcodes balance false taxonomic assignments in metabarcoding studies

Francesco Mugnai[1], Federica Costantini[1,2], Anne Chenuil[3], Michèle Leduc[4], José Miguel Gutiérrez Ortega[5] and Emese Meglécz[3]

[1] Department of Biological, Geological and Environmental Sciences (BiGeA), University of Bologna, Ravenna, Italy
[2] Consorzio Nazionale Interuniversitario per le Scienze del Mare (CoNISMa), Roma, Italy
[3] Aix Marseille Univ, Avignon Université, CNRS, IRD, IMBE, Marseille, France
[4] STARESO marine station, Calvi, Corse, France
[5] TAXON Estudios Ambientales S.L., Alcantarilla (Murcia), Spain

Corresponding authors
Francesco Mugnai,
checcomugna@gmail.com
Federica Costantini,
federica.costantini@unibo.it

## ABSTRACT

**Background.** In metabarcoding analyses, the taxonomic assignment is crucial to place sequencing data in biological and ecological contexts. This fundamental step depends on a reference database, which should have a good taxonomic coverage to avoid unassigned sequences. However, this goal is rarely achieved in many geographic regions and for several taxonomic groups. On the other hand, more is not necessarily better, as sequences in reference databases belonging to taxonomic groups out of the studied region/environment context might lead to false assignments.

**Methods.** We investigated the effect of using several subsets of a cytochrome c oxidase subunit I (COI) reference database on taxonomic assignment. Published metabarcoding sequences from the Mediterranean Sea were assigned to taxa using COInr, which is a comprehensive, non-redundant and recent database of COI sequences obtained both from BOLD and NCBI, and two of its subsets: (i) all sequences except insects (COInr-WO-Insecta), which represent the overwhelming majority of COInr database, but are irrelevant for marine samples, and (ii) all sequences from taxonomic families present in the Mediterranean Sea (COInr-Med). Four different algorithms for taxonomic assignment were employed in parallel to evaluate differences in their output and data consistency.

**Results.** The reduction of the database to more specific custom subsets increased the number of unassigned sequences. Nevertheless, since most of them were incorrectly assigned by the less specific databases, this is a positive outcome. Moreover, the taxonomic resolution (the lowest taxonomic level to which a sequence is attributed) of several sequences tended to increase when using customized databases. These findings clearly indicated the need for customized databases adapted to each study. However, the very high proportion of unassigned sequences points to the need to enrich the local database with new barcodes specifically obtained from the studied region and/or taxonomic group. Including novel local barcodes to the COI database proved to be very profitable: by adding only 116 new barcodes sequenced in our laboratory, thus

increasing the reference database by only 0.04%, we were able to improve the resolution for ca. 0.6–1% of the Amplicon Sequence Variants (ASVs).

## INTRODUCTION

Metabarcoding became a well-established technique applicable to a wide range of studies (*Taberlet et al., 2018*; *Slatko, Gardner & Ausubel, 2018*; *Ruppert, Kline & Rahman, 2019*). It can be applied in different fields such as diet analyses or interaction networks, but its primary field of application is the biodiversity assessment (*Compson et al., 2020*).

One of the challenges of metabarcoding is the production of robust datasets, that needs to be addressed both by appropriate experimental design, such as the use of replicates, mock communities and negative controls, the use of different markers and careful selection of primers (*Cristescu & Hebert, 2018*; *Alberdi et al., 2019*; *Zinger et al., 2019*; *van der Loos & Nijland, 2021*), coupled with appropriate bioinformatics pipelines adapted to the study design (*Zinger et al., 2019*). This step leads to a set of validated MOTUs (Molecular Operational Taxonomic Units) or ASVs (Amplicon Sequence Variants) in each sample, ideally free from artefacts like PCR or sequencing errors, chimeras, mis-tagging, or pseudogenes. The second challenge is the description and comparison of the biodiversity based on the outcome of the validated MOTUs or ASVs. This can be achieved either by assigning each ASV to a taxon, or the use of a blind approach where taxon richness and biodiversity are estimated from the number of MOTUs without the need of assigning them to taxa (*Nugent & Adamowicz, 2020*; *Marques et al., 2020*). This second option can be particularly appropriate for studies targeting geographical regions, or taxa where reference databases are highly incomplete (*Marques et al., 2020*), such as the marine environment (*Mugnai et al., 2021*) or for markers with low unspecific amplifications (*Collins et al., 2019*). However, knowing the taxonomic origin of the MOTUs or ASVs can be valuable if the aim is detecting specific groups of organisms or linking the community to ecosystem functioning, where this approach is often necessary for decisions on conservation measures. That said, in this study we will focus on the taxonomic assignment of ASVs, and the effect of the reference database used for the taxonomic assignment.

Metabarcoding can be based on one or several markers or even a complete mitochondrial or chloroplast genomes obtained by genome skimming (*Coissac et al., 2016*). The selection of the molecular markers for each study depends on several factors, including the variability of each marker in the target taxa, and the availability of the reference database(s), if the ASVs are needed to be assigned to taxa. Indeed, all taxonomic assignment methods, whatever their algorithms, depend on a reference database. An ideal reference database should be taxonomically diverse with a good coverage of each target group and should be free from mislabeled sequences. However, it is not clear whether a specialized database containing only sequences of the target taxonomic groups and geographical region is preferable to

a generalist database. Most studies that evaluated reference databases have compared databases of different origin (*Kocher et al., 2017*; *Park & Won, 2018*) or evaluated the precision of one given database (*Richardson et al., 2018*). Nevertheless, to our knowledge, it has not been tested how the reduction of a large generalist database, to smaller, more specialized databases tailored to the taxonomic group and/or geographic region affect the taxonomic assignment of environmental DNA (eDNA) sequences. Databases limited to a target taxon and/or region (*Sato et al., 2018*; *Arranz et al., 2020*; *Collins et al., 2021*; *Magoga et al., 2022*) are easier to create, smaller in size, and thus also easier to curate. However, they should be generated separately for each taxonomic group and updating them is usually not easy. Creating specific databases can be therefore a considerable bottleneck in metabarcoding studies. It is therefore tempting to use a generalized database, that contains up to date sequences of a given marker, irrespective of their taxonomic origin since even a species absent from the focal environment can in theory improve taxonomic assignment at genus or higher taxonomic level. However, sequences from taxa not present in the studied environment can also lead to false assignments if the target group is insufficiently covered (*Richardson et al., 2018*). In this study, we investigated the effect of using specific or generalist databases on the taxonomic assignments using four different algorithms for taxonomic assignment.

For Eukaryotes, the most frequently used markers for metabarcoding are the ribosomal RNA genes (16S, 12S, 18S), the Cytochrome Oxidase C subunit I (COI) gene and internal transcribed spacer sequences (ITS) (*Creer et al., 2016*; *Porter & Hajibabaei, 2018*). COI is the most sequenced marker for animal taxa (*Andújar et al., 2018*). Although it became clear that the COI gene is not necessarily sufficient to differentiate species in all animal groups (*Meier et al., 2006*; *Rubinoff, Cameron & Will, 2006*; *Roe & Sperling, 2007*), it still remains a widely used molecular marker for studies when taxonomic assignment of the ASVs is important, because its taxonomic resolution is the highest and its database is the most complete for most animal species. We have therefore chosen the COI marker for this study to assess one of the most challenging environments for biodiversity studies, the marine benthic invertebrate fauna, where the high presence of cryptic species (*Dennis & Aldhous, 2004*) needs molecular-based approaches for their identification (*Carvalho et al., 2019*). Albeit their importance towards ecosystem goods and service provisions is well known (*Guidetti & Danovaro, 2018*), reference databases for many marine taxa are still very incomplete (*Weigand et al., 2019*; *Duarte et al., 2021*; *Mugnai et al., 2021*), and efforts are thus needed. The high proportion of missing taxa in existing reference databases leads to many unassigned sequences. Although it is possible that many of the unassigned sequences are not necessary from the target group (for example COI primers can amplify algae, diatoms, bacteria and not just animals), it is likely that many sequences are unassigned by lack of appropriate reference sequences. Therefore, it is important to compete databases by vouchering of not yet barcoded species in community biodiversity assessments and submitting novel reference sequences to public databases.

In this article, we examine the repercussion on the taxonomic assignment of the use of generalist vs taxonomically and geographically refined databases and the effect of adding new barcoding sequences to a database.

## MATERIALS & METHODS

### New barcode sequences from the Mediterranean Sea

Within the framework of the co-founded ERA-NET MarTERA SEAMoBB (Solutions for Semi-Automated Monitoring of Benthic Biodiversity, https://seamobb.osupytheas.fr) project, 27 ARMS (Autonomous Reef Monitoring Structures, http://www.oceanarms.org) were placed on the rocky benthic substrate in several locations along the Mediterranean Sea between 15 m and 30 m of depth in July 2018 and recovered after one year (see Data S1). Macrofaunal organisms (>2 mm) were collected from the ARMS, individuals were sorted and morphologically identified. Due to taxonomic expertise, most of the specimens of polychaetes and crustaceans were identified in Italy (Ravenna), while gastropods and echinoderms were identified in Spain (Murcia) and France (Calvi). Individuals were fixed in molecular grade absolute ethanol, and then sent to the Ecological and Environmental Genetics Laboratory (GEA laboratory) of the University of Bologna in Ravenna, Italy, to be barcoded for the COI marker.

Field sampling authorizations were released in France from the Direction Interrégionale de la Mer Méditerranée (D.I.R.M), Préfecture de Corse and région PACA (Arrêté no 901 du 20 décembre 2017 and Arrêté no. 897 du 17 décembre 2018). Similarly, field sampling authorizations from Spain were released in the Cabo de Palos Marine Reserve from the Secretaría General de Pesca of the central administration (Authorization 2/18) and from the Servicio de Pesca y Acuicultura of the regional administration (Ref. 1/19). Field sampling activities in Italian sampling sites were not needed, as no MPAs (Marine Protected Areas) were involved in field sampling activities.

DNA extraction was performed on each organism separately using E.Z.N.A. Mollusc DNA kit (Omega Bio-tek Inc., Norcross, GA, USA), following the protocol provided by the manufacturer. PCR amplification of the COI barcode was performed using jgLCO1490 (forward) and jgHCO2198 (reverse) degenerated primers (5′-TITCIACIAAYCAYAARGAYATTGG-3′ and 5′-TAIACYTCIGGRTGICCRAARAAYCA -3′ respectively) (*Geller et al., 2013*) amplifying the "Folmer region". A final PCR volume of 13 µl was obtained, with 6.5 µl of AmpliTaq Gold 360 Mastermix (Thermo Fisher Scientific, Inc., Waltham, MA, USA), 0.5 µl of each forward and reverse primer (10 µM working concentration), 4.5 µl of PCR-grade water and 1 µl of DNA template. If necessary, DNA template and PCR-grade water volumes were adjusted to increase PCR yield. The PCR protocol included 1 min of initial denaturation at 95 °C, 35 cycles with 30 s of denaturation (95 °C), 45 s of annealing (47 °C) and 1 min of elongation (72 °C). A final 5 min extension at 72 °C was added. Amplified fragments were purified using the ExoSAP-IT Express kit (Thermo Fisher Scientific, Inc., Waltham, MA, USA) following the manufacturer's recommendations, and sent to Macrogen Europe BV (Macrogen Inc., Amsterdam, The Netherlands) for Sanger sequencing. MEGA X software (*Stecher, Tamura & Kumar, 2020*) was used to clean, align, and analyze the barcoded sequences, and the taxassign command of VTAM (version 0.2.0) (*González et al., 2020*) against the COInr database (*Meglécz, 2022a*) was used to detect potential contaminants or mislabeled sequences. The newly obtained

barcodes were submitted to GenBank (accession numbers ON716004–ON716119) and the list of the taxa with their metadata are shown in Data S2.

## Test dataset of ASVs

ASV sequences published by (*Wangensteen et al., 2018*) (https://peerj.com/articles/4705/#supp-12) were used as a test dataset. This same set of ASVs were assigned to taxa using generalist or specific databases (see description in the Reference Databases section), to investigate the effect of the database on the precision of the taxonomic assignments. Since we intended to evaluate the effect of adding to the database new barcodes from the Mediterranean Sea, we have selected only the ASVs present in Mediterranean samples. This led to a set of 7,179 ASVs coming from samples of 25× 25 cm natural substrate scraping from the Mediterranean (Data S3). The mentioned samples from Wangensteen and colleagues (2018) were amplified using the mlCOIintF-XT forward primer (5′-GGWACWRGWTGRACWITITAYCCYCC-3′ *Wangensteen et al., 2018*) and jgHCO2198 reverse primer (5′-TAIACYTCIGGRTGICCRAARAAYCA-3′ *Geller et al., 2013*), amplifying the ca. 313 bp "Leray fragment".

## Reference databases

Four different reference databases were used to assign the sequences of the ASVs of same test dataset to taxa. The COInr database (*Meglécz, 2022a*; *Meglécz, 2022b*) was chosen as global database, since it is a recent and comprehensive database of COI barcoding sequences, containing sequences from the NCBI-nt and BOLD databases, which are the major repositories of COI sequences. COInr contains sequences of all available taxa with COI sequences in the source databases except for sequences assigned to environmental samples (*e.g.,* taxID:100272, uncultured eukaryote). The size of COInr is reduced by a taxonomically aware dereplication algorithm (*Meglécz, 2022a*).

Three customized databases were created from COInr. First, a COInr-WO-Insecta database was generated from COInr, with all insect sequences removed as this class is not expected in (subtidal) marine samples. Most sequences in the COInr reference database belong to Insecta, which can lead to misassignments of ASVs. This database represents a very simple but considerable reduction of the database, without the need of a species or higher-level taxon list specific to the studied region which can be difficult to obtain. Since insects are not expected in the marine (subtidal) environment, all assignment to this class can be unambiguously regarded as false allowing an easy detection of these cases. Then, a list of Mediterranean families (Data S4) was retrieved from OBIS (*OBIS, 2022*, http://www.obis.org), in order to create a reference database focused on the geographic area of interest, the Mediterranean Sea LME (Large Marine Ecosystem). All sequences belonging to these families in the COInr-WO-Insecta database were used to create the COInr-Med database. Retaining all sequences of Mediterranean taxonomic families implies keeping also sequences from genera or species which are not present Mediterranean Sea according to OBIS. This allowed to account for some true Mediterranean species missing from the OBIS list. Furthermore, sequences from non-Mediterranean species in the database can serve to obtain an assignment at the genus or family level. Finally, we added our custom COI

barcodes (Data S2) to the COInr-Med database, generating the COInr-Med+ database. From each of the four datasets, sequences that covered at least 80% of the COI region amplified by the Leray primers (*Geller et al., 2013*) were selected and trimmed and these databases were formatted for VTAM (version 0.2.0) (*González et al., 2020*), RDP Classifier (version 2.13) (*Wang et al., 2007*) and QIIME2 (version Core 2022.2) (*Bolyen et al., 2019*) programs, that were used for taxonomic assignments. All sequence selection, trimming and formatting the databases were done by mkCOInr (version 0.2.0) (*Meglécz, 2022a*; *Meglécz, 2022c*) and the full list of commands used to obtain the three customized databases from COInr can be found in Data S5.

### Taxonomic assignment methods

All 7,179 ASVs of the test dataset were assigned to taxa, using four different algorithms and the four reference databases (Fig. 1).

Among the four taxonomic assignment algorithms, QIIME2's (version Core 2022.2) classify-consensus-blast, (*Bokulich et al., 2018*, referred to as QIIME_BLAST hereafter) and the taxassign function implemented in VTAM v−0.2.0 (*González et al., 2020*, VTAM hereafter) are both alignment-based methods using BLAST, but the algorithms are different. By default, the QIIME_BLAST algorithm identifies the lowest taxonomic group (LTG) that includes at least 51% of the ten best BLAST hits with at least 80% of identity. We ran this algorithm using three different identity thresholds (97%, 90% and 80%) on all four databases.

The other BLAST-based taxonomic assignment was VTAM's taxassign function (*González et al., 2020*). While QIIME_BLAST algorithm needs a fix similarity threshold, which is arbitrary and can rarely be applied to all sequences, VTAM's taxassign command uses a series of different identity thresholds sequentially (100%, 99%, 97%, 95%, 90%, 85%, 80%, 75%, 70%), and stops at the first identity threshold when an LTG can be established. For each identity threshold, the LTG is determined as the smallest taxon that contains 90% of the best hits (covering at least 80% of the amplicons length). Below the 97% identity threshold a further condition should be met: the LTG is established only if at least three different taxa are found among the best hits. Since the algorithm stops at the highest percentage of identity threshold where an LTG could be established, each assignment is associated with an identity threshold: the highest it is, the more likely it is to be correct. Assignments under 80% are highly unreliable, therefore, we have ignored them in this manuscript.

Both QIIME2's classify-sklearn algorithm (*Bokulich et al., 2018*), QIIME_SKLEARN, hereafter) and RDP classifier (version 2.13, *Wang et al., 2007* RDP, hereafter) are multinomial naïve Bayes classifiers, based on the k-mer compositions of the taxa in the training dataset and the sequence to be assigned. Both algorithms were run using default parameters with all four databases, and assignments with lower than 70% of confidence value were ignored.

The taxonomic assignment attributes a taxon name to each ASV. Each assignment is characterized by the taxonomic lineage of the taxon, and resolution or taxonomic rank (unassigned, phylum, class, order, family, genus species). The resolution increases from
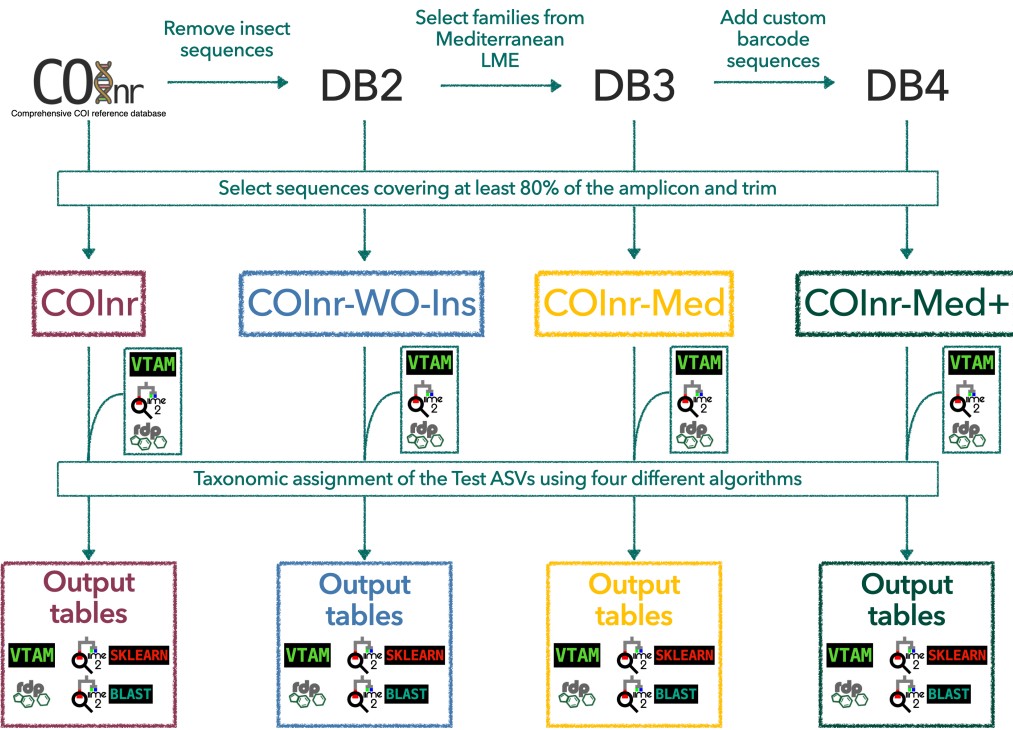

**Figure 1** **General outline of the workflow to generate the custom reference databases, to format them for the different taxonomic assignment algorithms, and to obtain the final output tables.** COInr: COI barcoding sequences from NCBI and BOLD, COInr-WO-Ins: COInr without insect sequences, COInr-Med: Sequences of COInr-WO-Ins, from a list of families present in the Mediterranean Sea, COInr-Med+: COInr-Med enriched with 116 new barcodes from the Mediterranean Sea. Taxonomic assignment methods: VTAM (VTAM taxassign), RDP (RDP_classifier), QIIME2_sklearn(QIIME2 classify-sklearn) QIIME2_BLAST (QIIME2 classify-consensus-blast with 80% similarity threshold).

unassigned to species. The taxonomic lineage is defined as a series of nodes in a hierarchical structure that connect the taxon to the top of the hierarchy (root or LUCA) (*Sakamoto & Ortega, 2021*).

The taxonomic assignments of the test dataset obtained by using the four reference databases were compared two-by-two: COInr *vs.* COInr-WO-Insecta, COInr-WO-Insecta *vs.* COInr-Med, COInr-Med *vs.* COInr-Med+. Particularly, we were interested in detecting (i) overall changes in taxonomic resolution, (ii) de novo assignments, and (iii) loss of assignments (sequences previously assigned to a taxon that became unassigned when using another database).

# RESULTS

## Overall taxonomic resolution

The identity threshold used for the QIIME_BLAST had a profound effect on the number of sequences assigned to taxa and on the resolution of the taxonomic assignment. A very stringent 97% identity threshold left ca. 86% of the sequences unassigned, but 75–87%

of the remaining sequences are assigned to species or genus. On the other extreme, the number of unassigned sequences was seriously reduced using the 80% identity threshold, leaving only 30–34% of the sequences unassigned, but the resolution drastically decreased, since only 36–50% of the other sequences are assigned to a species or genus level (Fig. S1, Data S6–S9). Since the high identity threshold seems to be more adapted to studies where the target taxa are very well covered in the reference database, and this is clearly not the case in our study on marine invertebrates, we chose the 80% identity threshold for further analyses, which is the default value of the QIIME_BLAST algorithm.

After performing the taxonomic assignment with all four databases and four algorithms, the resolution level of the assigned ASVs varied from phylum to species level, with strong variations of the number of ASVs assigned to a given taxonomic level across assignment methods (Fig. 2). However, when comparing the proportions of ASVs assigned to a given taxonomic level across the four different databases, all four assignment methods revealed the following trends: For almost all methods and databases (except for the COInr and the COInr-WO-Insecta databases with QIIME_SKLEARN algorithm) the unassigned category was the most frequent. The number of unassigned sequences varied strongly, being the lowest for the QIIME_SKLEARN (ranging from 1,345 to 2,155 among databases) and highest in VTAM (ranging from 3,390 to 3,956) methods, respectively. As expected, removing sequences from the COInr database (COInr-WO-Insecta, COInr-Med) increased the number of unassigned ASVs across all methods (Fig. 2). At the same time, the number of assignments at low resolution levels (phylum-order) tended to decrease, and this tendency was most remarkable in the QIIME_SKLEARN algorithm. The number of ASVs assigned to a high resolution level (genus-species) increased most particularly when reducing the database to the Mediterranean families (*i.e.*, using the COInr-Med databases with respect to the COInr-WO-Insecta) (Fig. 2).

When adding new barcodes to the Mediterranean database (COInr-Med+), the most noticeable change was the increase of the number of sequences assigned to species (from 20 to 54 additional ASVs, according to the algorithm). (Data S6, S9–S12).

## Pairwise comparisons of databases

Figure 3 represent the proportion of ASVs where the assignment is different between two databases. These changes can be compatible, meaning that the two lineages are the same, but the resolution of the assignment has changed (*e.g.*, one assignment is to the species *Idmidronea atlantica* the other is the *Idmidronea* genus), or incompatible, where there is a contradiction between lineages (*e.g.*, Bryozoa, Gymnolaemata *vs.* Bryozoa, Stenolaemata).

The proportion of ASVs where the assignment changed between different databases was particularly high for the two algorithms implemented in QIIME2 reaching 67% for the QIIME_SKLEARN algorithm between the COInr-WO-Insecta and the COInr-Med databases. Although the proportion of compatible changes was higher than the incompatible ones in all comparisons, the proportion of incompatible changes still reached 8−9.5% for the QIIME_BLAST and 10.8% in the QIIME_SKLEARN algorithms (Data S13).

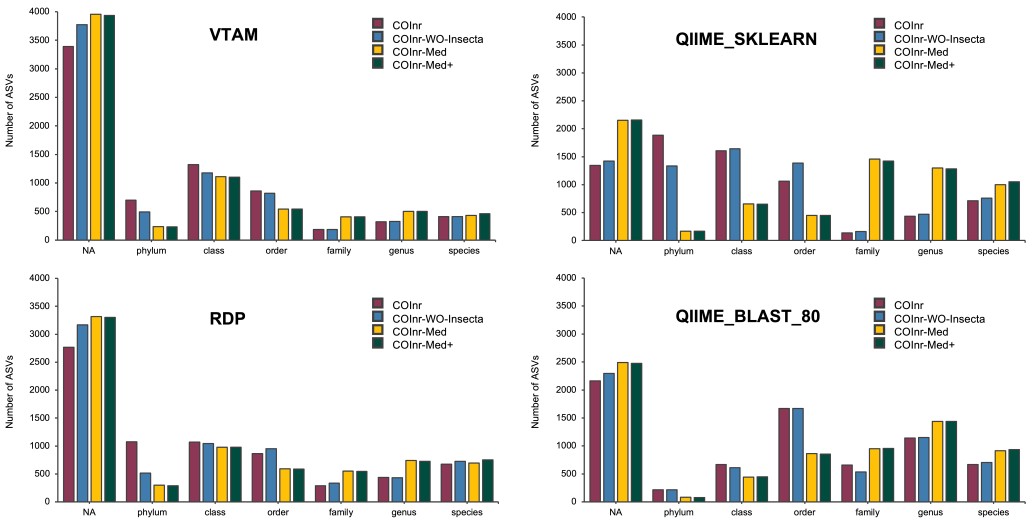

**Figure 2** **Taxonomic resolutions of ASVs using different algorithms for taxonomic assignment.** VTAM (VTAM taxassign), RDP (RDP_classifier), QIIME_sklearn (QIIME2 classify-sklearn) QIIME_BLAST_80 (QIIME2 classify-consensus-blast with 80% similarity threshold).

## The effect of removing irrelevant sequences from the reference database on the resolution

Figure 4 represents the number of ASVs with increased and decreased resolution of their taxonomic assignments between two databases. Most of the observed changes of taxonomic assignment resulted in compatible lineages, but the number of assignments with incompatible lineages was not negligeable for the two QIIME2 algorithms (Fig. 4 and Fig. S2, Data S13).

The COInr-WO-Insecta database contained only 764,257 sequences compared to the 3,121,590 of COInr, thus insect sequences represented 75.5% of the COInr database.

When removing the high number of irrelevant insect reference sequences from COInr we expected the elimination of false assignments to insects or arthropods. Indeed, when comparing the assignments using the COInr and the COInr-WO-Insecta databases, many sequences became unassigned (488, 570, 522, 265) with all four methods, most of which has been previously assigned to insects (304, 46, 79, 251), or for Arthropoda without further precision (141, 497, 415, 8) for VTAM, RDP, QIIME_SKLEARN and QIIME_BLAST, respectively. In addition, most ASVs that switched to unassigned had previously a low-resolution assignment (phylum, class, order, Fig. S2, Data S9–S13). On the other hand, the removal of insect sequences also helped in increasing the taxonomic resolution of ASVs (Fig. 4) and most of them were assigned to Arthropoda (206/217, 143/352, 532/653, 320/357) with the COInr database, for VTAM, RDP, QIIME_SKLEARN and QIIME_BLAST, respectively (Fig. S2, Data S9-S13).

The COInr-Med database had 293,580 sequences, representing a 61.6% of reduction of the database size compared to COInr-WO-Insecta. Switching from the COInr-WO-Insecta to the COInr-Med database, the increasing resolution was the most frequent type of change

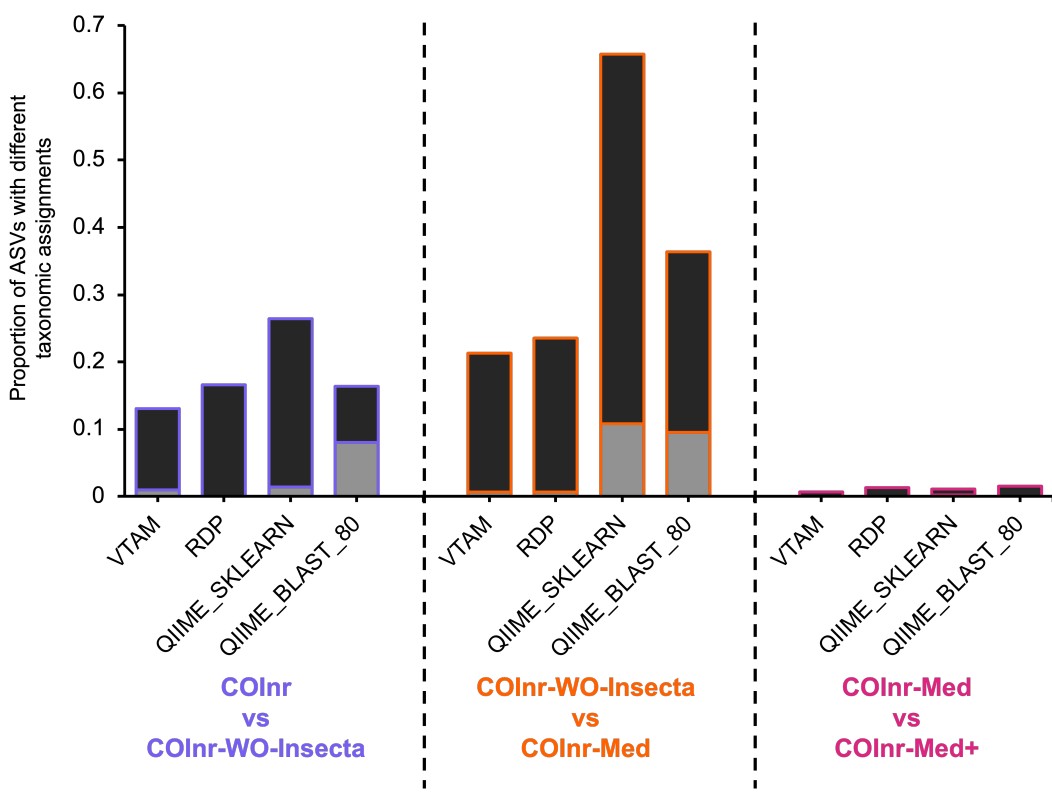

**Figure 3  Proportion of ASVs where the taxonomic assignments were different when comparing two different databases.** Black: compatible changes (same lineage but different resolution). Grey: incompatible changes.

(598, 828, 2321, 1397), followed by ASVs becoming unassigned (457, 329, 1372, 469), for VTAM, RDP, QIIME_SKLEARN and QIIME_BLAST, respectively (Fig. 4, Data S13).

## Adding new barcodes from the local fauna to the reference database

Out of the 116 newly barcoded specimens, 96 were identified to a species, 17 to genus, one to family and two to class level. Among the barcodes, 52 sequences belonged to taxa (45 different species and 1 genus) not yet present in the COInr database, 61 were from taxa (one class, one family, five genera, 34 species) already present in COInr but different from already existing sequences due to intra-taxon variability, and 3 were identical to reference sequences of the same taxon. The 52 sequences from taxa new to COnr database are coming from six different phyla: Annelida (19), Arthropoda (9), Bryozoa (3), Echinodermata (2), Mollusca (18) and Platyhelminthes (1). The complete list of sequences, lineages and NCBI accessions can be found in Data S2.

When adding 116 local barcoding sequences (COInr-Med+) to the COInr-Med database (Fig. 4C, Fig. S2) the assignment of few sequences changed but the increase of resolution clearly outnumbered the ASVs with decreased resolution (Fig. 4). The total number of assignments that changed resolution was 48, 97, 76 and 107 for VTAM, RDP, QIIME_SKLEARN and QIIME_BLAST, respectively (Fig. 4, Data S13). This represents

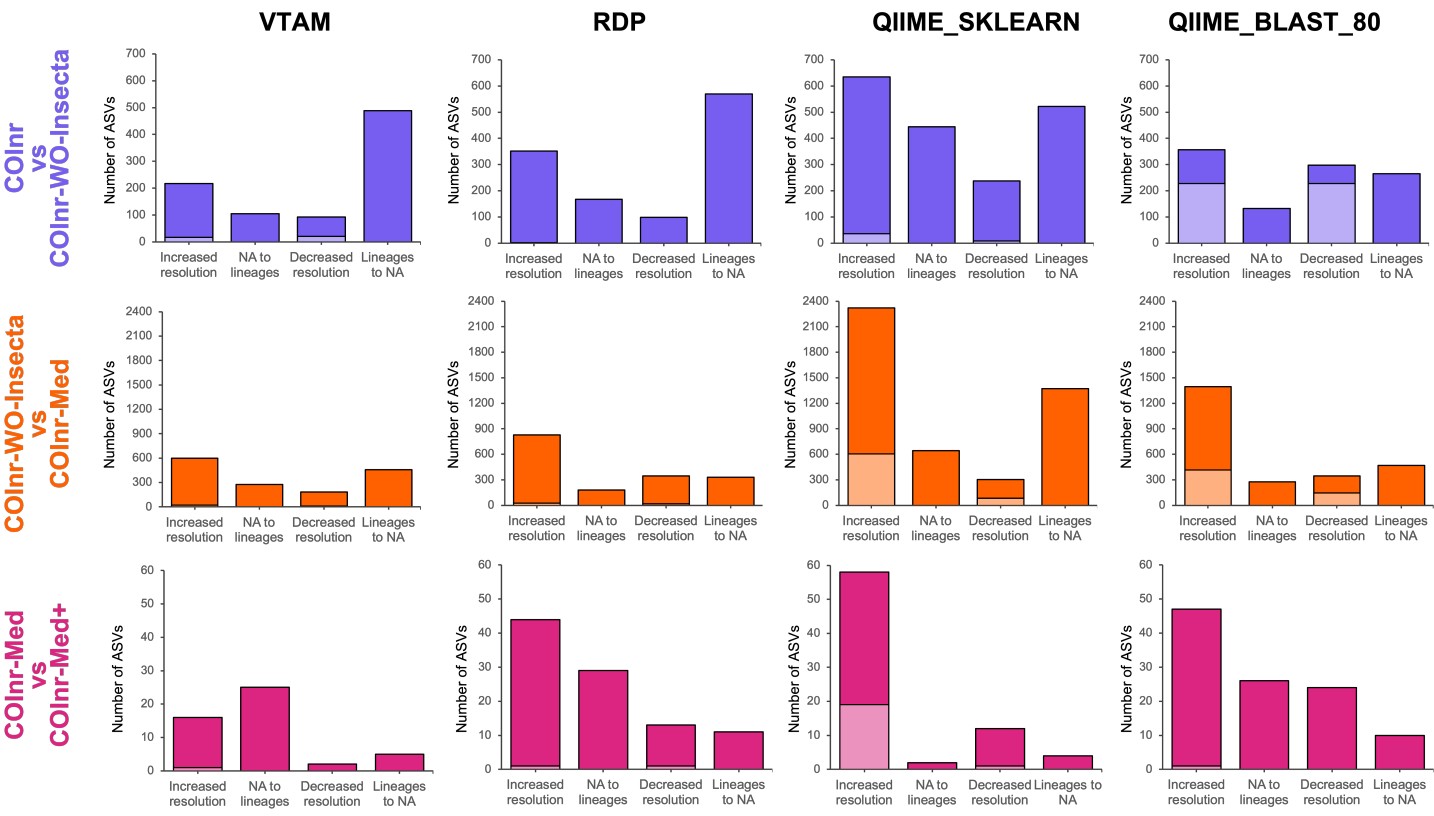

**Figure 4** **Overall changes in taxonomic resolution between taxonomic assignments of two databases.** Compatible changes are in bright colors, incompatible changes are in light colors.

0.7−1.4% of the ASVs, while the 116 new barcodes make up to only 0.04% of the COInr-Med database. As expected, most changes corresponded to increased resolution, meaning these ASVs resembled more to a new barcode than sequences in the COInr-Med database. However, 6, 24, 16 and 26 sequences (for VTAM, RDP, QIIME_SKLEARN, respectively) had decreased resolution which is likely the result of the presence of mislabeled sequences in the database.

## DISCUSSION

The use of a generalist database including all available taxa seems an evident solution for taxonomic assignment, as the presence of non-relevant taxa in the database should have no effect on the output. This would likely be the case if the generalist database were almost complete, had good taxonomic coverage of the "tree of life," or at the least, had great coverage of the target taxa in the geographical region under study. However, this is an ideal situation very far from the reality for COI data in general (*Weigand et al., 2019*) but especially in marine environments (*Wangensteen et al., 2018*; *Mugnai et al., 2021*). The incompleteness of the reference databases was also reflected indirectly by our analyses, since a non-negligeable proportion of the assignments changed when removing

irrelevant sequences from the database (Fig. 3). This proportion was particularly high in QIIME_SKLEARN and QIIME_BLAST assignments and even most importantly, the proportion of incompatible assignments were non-negligible for these methods, indicating that they are more sensitive to the database content and more likely to produce false assignments with a generalist database than the other two more conservative methods.

The number of COI sequences available for different phyla is extremely variable, with strong overrepresentations of Arthropoda (*Meglécz, 2022a*) and particularly insects, and substantially lower coverage for many marine taxa (*Weigand et al., 2019*; *Mugnai et al., 2021*). This was clearly evidenced by our findings, where roughly 20–50% of the tested marine ASVs could not be assigned to taxa with great variations among taxonomic assignment methods and global or specialized databases. The insects are by far the most represented class in the COInr database (*Meglécz, 2022a*) but these sequences are irrelevant in marine metabarcoding studies. The low coverage of marine taxa, coupled with the presence of many insect sequences in the database, resulted in the false assignment of many ASVs to insects, which are clearly false positives, or to the Arthropoda phylum, which is also likely to be incorrect in many cases. The obvious and simple solution seems to remove insect sequences from the reference database which do not need a taxon list specific to the studied region. As expected, this led to the increase of the number of unassigned sequences, as most of them were previously assigned to Arthropoda and particularly to insects. Therefore, the overall taxonomic assignment became more accurate by eliminating clearly incorrect assignments. Furthermore, removing insect sequences from the database allowed for a higher resolution assignment of non-negligible proportion of ASVs (Fig. 4). This increase should be considered with caution since it can happen in two ways. First, the new assignments (WO insecta) can be essentially based on sequences that have a similarity comparable to the ASVs as some of the insect sequences, therefore the new assignment is probably not very robust, especially for a simple best-blast hit approach (QIIME_BLAST). However, we observed increased resolution with all four methods, which gives more credit for this observation. Second, the resolution can increase if mislabeled sequences have been eliminated from the database, which is a sign of improvement. Since 75.5% of the COInr sequences have been eliminated from the COInrWO-Insecta database and mislabeling is a known problem in the NCBI-nt and BOLD databases (*Bidartondo, 2008*; *Meiklejohn, Damaso & Robertson, 2019*), which are the source databases for COInr, it is plausible to think that some of the eliminated sequences has been mislabeled.

Following the same line of logic, restricting the reference databases to species present in a specific region of study is clearly a good direction. However, obtaining a complete list of species for a taxonomically diverse environment is not necessarily simple since species inventories are likely to be incomplete. Due to the low taxonomic coverage of the database for marine taxa, limiting the reference database to a local (potentially incomplete) species list can lead to a very limited reference database, and an extremely high number of unassigned ASVs. This loss can be limited by including all sequences of genera or families of the studied region, even for species that have not been reported as present in the region. Reducing the database from COInr-WO-Insecta to COInr-Med, again allowed to obtain a higher taxonomic resolution for many ASVs at the price of a further increase of the

 

number of unassigned sequences. Due to the incompleteness of the OBIS database, it is difficult to affirm that all ASVs that became unassigned from the COInr-WO-Insecta to the COInr-Med databases had previously an incorrect assignment. However, most of these ASVs had only a phylum level resolution with COInr-WO-Insecta (Fig. S2), indicating a lack of reference sequences sufficiently similar to these ASVs and leading to assignments with little information, most of which are based on irrelevant sequences.

Our databases have not gone through extensive curation, and it is well known that its source databases, especially the NCBI-nt database, contain a high number of mislabeled sequences (*Bidartondo, 2008*; *Meiklejohn, Damaso & Robertson, 2019*). Unfortunately, semi-automatic methods to detect mislabeled sequences (*Kozlov et al., 2016*; *Rulik et al., 2017*; *Meiklejohn, Damaso & Robertson, 2019*) are not adapted for the curation of databases of hundreds of thousands of sequences, and even our smallest database (COInr-Med) is still too large for automatized curation. Mislabeled sequences can either lead to incorrect assignment, especially if the true taxon of the query sequence is not represented by other correctly labelled sequences in the database, or low-resolution assignment if they are present. Therefore, size of the database permitting, a curation of databases is highly desirable. This becomes feasible for local, small case studies with limited taxonomic scope (*Kocher et al., 2017*; *Collins et al., 2021*).

Although it is reasonable to think that the taxonomic assignment of the ASVs became more robust using a local database limited to Mediterranean families, the proportion of unassigned ASVs is still worryingly high. This is partially the result of the observation that COI primers can amplify a large variety of organism including plants, fungi and even bacteria. Thus, amplicons are not limited to animals or a particular taxa the study is focusing on (*Collins et al., 2019*). Since most effort of COI barcoding concentrate to animal species, non-metazoans are underrepresented in COI reference libraries and may represent a significant part of the unassigned taxa. At the same time, it is also clear that many COI barcoding databases are incomplete for animal taxa and this calls for an intensive effort to couple metabarcoding studies with barcoding of the local fauna/flora. This barcoding should not be necessarily limited to the COI markers. COI showed lower reproducibility than 12S ribosomal RNA gene in aquatic environment but 12S lacks adequate references (*Collins et al., 2019*). To make the most out of the existing COI sequences and to generate new ones, and at the same time accumulate reference sequences of other mitochondrial sequences (including 12S), the genome skimming technique is now a feasible alternative (*Coissac et al., 2016*) that can be chosen as an alternative to COI barcoding with marginal extra cost, but with the advantage of obtaining whole cell organelle genomes.

Adding just 116 new barcodes to the COInr-Med database, which is negligeable compared to the size of COInr-Med (0.04%), still allowed to increase the resolution of 0.6% (VTAM), 1% (RDP), 0.8% (QIIME_SKLEARN) and 1% (QIIME_BLAST) of the ASVs, most often going from unassigned to species level. This is a clear improvement providing the good quality of the new barcodes. However, adding new barcodes also resulted in decreased resolution in a few cases, which is likely to be a sign of mislabeled sequences in the database and calls for database curation whenever it is possible.

## CONCLUSIONS

Our comparisons showed the importance of having reliable reference databases, filled with custom sequences, refined at a local study area and without the presence of unexpected taxa. Eliminating irrelevant reference sequences allows lighter and more adequate databases to be built that comes at the price of increasing the number of unassigned ASVs. However, this is not always a negative consequence. ASVs that became unassigned with a more specific database were generally assigned to a low-resolution taxonomic level and they had been very likely incorrectly assigned with the larger database. The use of a suitable database results in a cleaner and more accurate taxonomic identification.

Creating a local database specific to the study, however, cannot compensate for the need of barcoding to fill the gaps in the taxonomic coverage of databases which should be done in association with expert taxonomists. We tested only the COI marker, since it has most reference sequences for marine invertebrates. These findings are likely to hold even stronger for other markers or whole organellar genomes, with low taxonomic coverage.

## ACKNOWLEDGEMENTS

This work is part of Mugnai F. PhD project on "Monitoring of Mediterranean benthic biodiversity: an integrative taxonomic approach" at the Alma Mater Studiorum –Università di Bologna. This work was part of the European project SEAMoBB: "Solutions for Semi-Automated Monitoring of Benthic Biodiversity". Special thanks to Dr. Kenan O. Matterson for the field sampling and the molecular-related contributions, bioinformatics included. Thanks to Marine Biology MSc students involved in SEAMoBB project field- and lab-related activities during 2019 and 2020 campaigns. Thanks to Dr. Barbara Mikac for the identification of Polychaeta in Ravenna laboratory. Thanks to all the diving centers for field activities support (Rovinj Sub, Rovinji, Croatia; Palinuro Sub, Palinuro, Italy; Accademia Blu Diving Center, Livorno, Italy), and to the CIBM (Centro Interuniversitario di Biologia Marina ed Ecologia Applicata "G.Bacci", Livorno, Italy) and the Ruđer Bošković Institute (Rovinj, Croatia) for their support during field sample processing facilities. We acknowledge the scientific SCUBA-diving French facility of the OSU Pythéas for ARMS installation (Virgile Calvert, Sandrine Chenesseau, Dorian Guillemain, Anne Haguenauer, Frédéric Legendre, Christian Marschal, Laurent Vanbostal, Frédéric Zuberer) and Laetitia Plaisance, Marjorie Selva, Térence Legrand, Erwan Bouchereau, Florent Marschal, Pascal Mirleau, Vincent Rossi for their help with ARMS processing /sorting. We also want to thank the TAXON team for 2019 and 2020 SEAMoBB sampling campaigns participation in Murcia, Spain, Samanta Ortuño and Pedro Castaño for lab-related activities while for field sampling and the identification of fauna: Desiderio Andreo, Ezequiel Martínez (Crustacea, Cnidaria), Carlos Carrasco (Polychaeta) and Valentín Aliaga (Echinodermata and Mollusca). Thanks to the team of STARESO for collecting samples during the 2019 and 2020 SEAMoBB sampling campaigns and to Annick Donnay for the identification of selected individuals.

### Funding

This work was supported by the European project SEAMoBB (Solutions for sEmi-Automated Monitoring of Benthic Biodiversity), funded by ERA-Net Mar-TERA (id. 145) and managed by the ANR (Grant No. ANR-17-MART0001-02, P.I.: Anne Chenuil, Grant No. ANR-17-MART0001-03, PI: STARESO) and Ministero dell'Istruzione, dell'Università e della Ricerca (MIUR) Grant No. MIUR n. 728053 - SEAMoBB. The Spanish founder was CDTI, and the project name Soluciones Para El Monitoreo Semi-Automático De La Biodiversidad Bentónica with Id number SERA-20181031. The funders had no role in study design, data collection and analysis, decision to publish, or preparation of the manuscript.

### Grant Disclosures

The following grant information was disclosed by the authors:
European project SEAMoBB.
ERA-Net Mar-TERA: 145.
ANR: ANR-17-MART0001-02, ANR-17-MART0001-03.
Ministero dell'Istruzione, dell'Università e della Ricerca (MIUR): 728053.
CDTI.
Soluciones Para El Monitoreo Semi-Automático De La Biodiversidad Bentónica: SERA-20181031.

### Competing Interests

José Miguel Gutiérrez is employed by TAXON Estudios Ambientales S.L. as Joint Chief Executive Officer.

### Author Contributions

- Francesco Mugnai conceived and designed the experiments, performed the experiments, analyzed the data, prepared figures and/or tables, authored or reviewed drafts of the article, sampling efforts, sample processing, laboratory procedures, and approved the final draft.
- Federica Costantini performed the experiments, authored or reviewed drafts of the article, contributed reagents, materials, analysis tools; establishing sampling and laboratory protocols, and approved the final draft.
- Anne Chenuil performed the experiments, authored or reviewed drafts of the article, establishing sampling and laboratory protocols, and approved the final draft.
- Michèle Leduc performed the experiments, authored or reviewed drafts of the article, sampling efforts, sample processing, and approved the final draft.
- José Miguel Gutiérrez Ortega performed the experiments, authored or reviewed drafts of the article, sampling efforts, sample processing, and approved the final draft.
- Emese Meglécz conceived and designed the experiments, performed the experiments, analyzed the data, prepared figures and/or tables, authored or reviewed drafts of the article, sampling efforts, sample processing, and approved the final draft.
## Field Study Permissions

The following information was supplied relating to field study approvals (i.e., approving body and any reference numbers):

Authorization for field sampling in France, Arrêté n° 901 du 20 décembre 2017 and Arrêté n° 897 du 17 décembre 2018, were released from the Direction Interrégionale de la Mer Méditerranée (D.I.R.M), Préfecture de Corse and region PACA.

Authorizations for field sampling in Spain were released from the Secretaría General de Pesca of the central administration (Authorization 2/18) and from the Servicio de Pesca y Acuicultura of the regional administration (Ref. 1/19).

## DNA Deposition

The following information was supplied regarding the deposition of DNA sequences:

The new barcoding sequences are available at GenBank: ON716004 to ON716119.

## Data Availability

The data is available in the Supplementary Files.

## Supplemental Information

Supplemental information for this article can be found online at http://dx.doi.org/10.7717/peerj.14616#supplemental-information.

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
