# Peer review of "Be positive: customized reference databases and new, local barcodes balance false taxonomic assignments in metabarcoding studies"

_PeerJ, doi:10.7717/peerj.14616_

## Round 0.1 · original submission · Major Revisions

Two expert reviewers provided thoughtful and detailed comments on this manuscript, which could serve to highlight the importance of regional or local curated reference libraries and the importance of generating new data for reference databases, but requires major revisions to improve the quality and reproducibility of the science involved.

There is some question on why the study focused solely on the COI barcoding region, when metabarcoding often relies on a number of markers. A genome skimming approach could be used to simultaneously sequence a number of high-copy number markers for a marginal increase in sequencing cost. This is something that should be addressed.

The introduction to the paper is quite short and does not include numerous papers that have discussed or address this question of taxonomic assignments to metabarcodes. In the introduction, your research question should be clearly defined, and hypotheses presented (if any).

The methods require additional information on controls and blanks for assessing contamination, and primers for both metabarcoding and barcoding should be listed. Accession numbers for all sequences should be provided and accessible to assess sequence quality should the reviewers wish it.

The results focus primarily on effects of removing taxa to improve the database which is useful, but does not discuss taxonomy assignment quality and uses a potentially non-peer reviewed pipeline (VTAM), which like the reviewers, I could only find the preprint version, and no published version of this tool.

The figures are quite nice, but perhaps a new figure showing the workflow for refining reference databases and assigning taxonomies would be beneficial.

I believe this manuscript can be dramatically improved and provide a useful contribution to the metabarcoding literature if the authors address each of the comments from the reviewers.

Reviewer 1 ·

Excellent Review

This review has been rated excellent by staff (in the top 15% of reviews)
EDITOR COMMENT
The reviewer provided a very thorough and professional review of this manuscript, suggesting additional analyses and reference citations that will greatly improve the flow and application of the paper. Thank you very much for your efforts in reviewing this manuscript!

Basic reporting

The article does not contain enough introduction and background information. It does not fit into the broader field of knowledge of taxonomic labelling of eDNA. The work partially overlaps with Meglécz (doi:https://doi.org/10.1101/2022.05.18.492423) and does not seem to be a self-contained publication unit.

The referenced literature is biased and incomplete for such an important topic. For example, there are only 10 references in the introduction. Most of them have a limited focus (e.g. marine biodiversity, COI, specific databases). There is no reference related to the problem of taxonomic labelling of eDNA reads. They ignored important, relevant work. Here are only a few recent examples:

- Singer et al. (2019) Comprehensive biodiversity analysis via ultra-deep patterned flow cell technology: A case study of eDNA metabarcoding seawater. Sci. Rep. 9(1), 1-12.

- Marques et al. (2020) Blind assessment of vertebrate taxonomic diversity across spatial scales by clustering environmental DNA metabarcoding sequences. Ecography 43(12), 1779-1790.

- Nugent et al. (2020) Alignment-free classification of COI DNA barcode data with the Python package Alfie. Metabarcoding Metagenomics 4, e55815.

- Mathon et al. (2021) Benchmarking bioinformatic tools for fast and accurate eDNA metabarcoding species identification. Mol. Ecol. Resour. 21(7), 2565-2579.

Experimental design

General Comments

The manuscript evaluates database modifications to improve “robustness” of the taxonomic labelling of ASVs. Although it is addressing an important problem in eDNA Metabarcoding and AmpSeq the scope is rather narrow. The focus of improvement is solely on the reference database. The work focuses strongly on marine diversity and COI.

The authors suggest restricting existing and extensive databases to the expected diversity. The authors also suggest supplementing the databases with more local references. Both ideas are not necessarily new, but have the potential to improve the annotation of ASVs. Unfortunately, both ideas also have their disadvantages. These should be considered, especially considering using of a best-blast-hit approach. The authors argue mainly with quantity improvements, forgetting the importance of quality. A reference is not only dependent on its size but also on its diversity and accuracy of its contends. More assignments do not mean better assignments.

A third and simple improvement of reference databases to remove “bad” records (e.g. 33858 2836 no rank environmental samples) was not considered. An effective option implemented in the BOLD Identification System (IDS) search for COI.

The focus of the manuscript is on the reference database. The best-blast-hit approach with an identity threshold cutoff of 80% is not critically evaluated. They do not take other important aspect of improving annotation up. For example, the importance of primer selection (e.g. Leese et al. 2020), the use of different loci (e.g. Liu and Zhang 2021) or the use of new methods for annotation (Mächler et al. 2020 or Flück et al, 2022) remain unmentioned.

- Leese et al. (2020) Improved freshwater macroinvertebrate detection from eDNA through minimized non-target amplification
- Liu and Zhang (2021) Combining Multiple Markers in Environmental DNA Metabarcoding to Assess Deep-Sea Benthic Biodiversity.
- Mächler et al. (2020) Decision-making and best practices for taxonomy-free environmental DNA metabarcoding in biomonitoring using Hill numbers.
- Flück et al. (2022) Applying convolutional neural networks to speed up environmental DNA annotation in a highly diverse ecosystem.

The manuscript by Mugnai et al. in its current form has too little substance and would need to be expanded. A well-founded introduction of exiting approaches is missing. A critical examination of the approach used is missing. An in-depth discussion of further improvements is also missing. Reproducibility would have to be improved (e.g. version and parameter) and raw data (metabarcoding Illumina data) would have to be made available.

Specific Comments

- Barcoding from the Mediterranean Sea [94-129 / 186-190] - This section is an important part and a nice contribution to improved eDNA labelling. Unfortunately, it gets a bit lost. It was not immediately clear that this section refers to the “new barcodes”. Perhaps this section should be placed after the metabarcode part. In addition, a brief introduction would be useful. I also think the effort of the 116 new barcodes should be better highlighted and documented (possibly graphically).

- Missing Data [128-129] - I could not find the accession ON716004-ON716119 on NCBI GenBank.

- Metabarcoding test dataset [131-156] - The data processing and especially the taxonomic assignments are insufficiently described. There are many ambiguities that limit my assessment. I have no experience with VTAM and have to rely on the online manual and the publication. But here already lies the first major problem. I cannot find the publication [333] Gonzales et al. 2020 in Ecology. There is a preprint on BioRxiv, but the description is very superficial. Also, there are many optional processing steps and countless parameters and the data to process. So I don’t know how the data was processed and where the limits are. For example, I don’t know if ASVs with stop codons were filtered out and if the correct genetic code was used for this. I also don’t know if an abundance filter was used to reduce sequencing errors. It is not clear why the ASVs were additionally clustered with 97% identity. In fact, I don’t even know the version of VTAM that was used. The data preparation is a black box.

- Missing Data - Missing accession for submitted Illumina fastq paired-end raw read data.

- [138] It gives the impression that they used different primers for the new barcodes and the metabarcoding. Both primer pairs should be named so that it is clear that only the forward primer is different and that mlCOIintF/jgHCO2198 is a part of jgLCO1490/jgHCO2198.

- Metabarcoding test dataset [149-150] - Perhaps the most important aspect is the nuclear taxonomic assignment. The online VTAM manual for the taxassign command states that taxa name and rank are chosen based on lineages of the best hits. Whether a best-blast-hits (BBH) aligner performs better than a classifier (e.g. RDP classifier) would have to be discussed. My concern about the BBH approach is the susceptibility of erroneous assignments due to (few) erroneous references. Why an 80 percent identity threshold was used without considering the query coverage (the option is available in VTAM) is also not clear to me. The alignment length and the identity should be considered.

- Sample Design - Negative (and mock) samples are missing. There are no replicates. Suggested reading:

Sepulveda et al. (2020) The Elephant in the Lab (and Field): Contamination in Aquatic Environmental DNA Studies.

Stauffer et al. (2021) How many replicates to accurately estimate fish biodiversity using environmental DNA on coral reefs?

- Results [193-201] - Important information is missing. Statements such as “most sequences remained unassigned” are inaccurate. > We could not find taxonomic labels for 924 (57%) ASVs.

- Results [193-233] - When discussing the results, the limitation of the method used is completely ignored. Best-blast hits are used for the annotation and the resolution is based on it. How are several equally good hits evaluated? An improvement can only occur if a new added sequence gives a better hit than the best previous one.

Validity of the findings

- The authors work with absolute numbers and forget the (relative) abundance. If only half of the ASVs could be given a taxonomic label, this may not represent the majority of the data but only a small fraction.

- The work has a focused approach, and other approaches are completely ignored. Which makes the interpretation very one-sided. Here is a selection of papers that could help to close the knowledge gap:

Mächler et al. (2020) Decision-making and best practices for taxonomy-free environmental DNA metabarcoding in biomonitoring using Hill numbers.

Liu and Zhang (2021) Combining Multiple Markers in Environmental DNA Metabarcoding to Assess Deep-Sea Benthic Biodiversity.

Flück et al. (2022) Applying convolutional neural networks to speed up environmental DNA annotation in a highly diverse ecosystem.

Additional comments

Misconception: [67-68] “This step relies on reference databases, which need to be as complete as possible”. This statement by the authors is misleading and one-sided. The reference should certainly be as complete as possible, but it should also be of good quality and high diversity. Much does not help if it is wrong, especially if you use a best-blast-hit approach.

Supplemental file: Data_S5.html - It needs a decisive reference to the GitHub website where the Perl scripts can be found. https://github.com/meglecz/mkCOInr/tree/main/scripts

[246-247] - I find the statement “the insects are by far the most represented class in reference databases” problematic and certainly not true for all databases e.g. SILVA 16S or UNITE ITS.

Reviewer 2 ·

Excellent Review

This review has been rated excellent by staff (in the top 15% of reviews)
EDITOR COMMENT
This reviewer provided thoughtful and detailed suggestions for the authors to improve their manuscript and increase its utility in the metabarcoding world. The reviewer commented on data and code availability that could affect the reproducibility of results, and provided helpful comments on improving the flow of the draft. I would like to thank the reviewer for their efforts!

Basic reporting

This paper presents the argument of curating reference databases to local/regional taxonomic assemblages to increase the accuracy of taxonomic assignment of eDNA metabarcoding reads. The authors have successfully shown that by customizing the reference database to local taxa, read assignment improves in accuracy. The authors compare four databases; the COInr, COInr-WO-Insecta, COInr-Med, and COInr-Med+. The comparison of the last two databases shows that adding reference barcodes from local taxa can increase the number of ASVs assigned and assignment resolution. The serial reduction of the database to first remove insects and then remove all families not in the region has not yet been addressed in the literature. Rather, most studies focus on comparisons between a “global” database (here COInr) with a “local” database (here COInr-Med). Removing insects first, and then other non-local taxa seems redundant (as insects are also non-local taxa), and this should be collapsed into a single step that removes all non-local taxa (including insects). Keeping the comparison of COInr-Med with COInr-Med+, the addition of local barcodes, is interesting to illustrate the utility of additional barcodes. I recommend re-structuring the analyses, results, and discussion to exclude the serial removal of all insects and then other non-local taxa, and group that into all non-local taxa. Then the authors can add the additional focus of another assignment method as suggested in the next section.

The article is structured well with excellent figures and supporting data provided to enable the reader to fully understand the scope, analyses, and results.

There is no data deposition statement included although all barcodes have Genbank Accession Numbers. The Genbank sequences are not yet released so could not be checked.

Minor comments:
Line 56: change ‘leading’ to ‘making’.

Line 60-62: Settlement structures are a common way to assess invertebrate biodiversity, but not total marine biodiversity. The reference provided doesn’t support the statement that it is a preferred way to sample marine biodiversity. Please change the sentence to make it more accurate.

Lines 68-71: Please add in a comment on reference library completeness meaning more than just taxonomic completeness, more genes or mitochondrial genomes are also needed.

Lines 71-75: Whilst the taxonomic coverage in reference databases does need to increase in order to assign more reads to gain better interpretations for ecological questions, the majority of unassigned reads in COI metabarcoding datasets are non-metazoan. Please add a sentence to address this well-known issue in the introduction.

Line 82: This statement needs a citation or two.

Line 101: Please elaborate on which partners did the identification.

Line 103: Add comma after Italy.

Lines 187 – 190: Please increase accuracy of these statements. The sentence reads that some taxa were already present in COInr but different due to intraspecific variation, whereas Data S2 has additional existing taxa at the genus, family, and class levels.

Lines 228 – 233: Please explain the change in assignment for the 3 ASVs where taxonomic resolution decreased and why.

Line 241: Please re-structure this sentence for readability.

Experimental design

The description of the metabarcoding protocol did not include essential information to reproduce the study or evaluate the results. Please describe the positive and negative controls. Also please indicate what type of field and laboratory negative controls were used, and how these were used to remove potential contamination from your results. Positive controls can lead to contamination, unless they are non-organismal DNA that can be identified. All mention of controls in the methods text needs clarification. Please describe the molecular identification tagging system that was used with your fusion primers. For example, was the system designed to detect tag jumping?

VTAM software is shown in the literature cited section as being published in Ecology in 2020, but the DOI is the bioRxiv identifier and does not have an Ecology DOI. I cannot find evidence of this article being published in Ecology in 2020. If the pipeline hasn’t successfully undergone peer-review since being published as a preprint in 2020, I would question it’s use in the study and ask why the authors did not use a pipeline that has been published and widely cited such as QIIME. Further to transparency of VTAM, the preprint describes only a default value of 97% for the Lowest Taxonomic Group. I am assuming this is applied at the species level, and default percent identity values for interspecific variation at higher taxonomic levels are not shown in the preprint or described in the methods. The Data S3 file shows the range of ASVs assigned to genus level at 80% to 100% within all tested reference databases, whereas all species were assigned at the 97% and above level. Whilst these numbers may represent natural variation within the taxa, there is no acceptable justification to make the taxonomic assignment to genus at such variable levels of percent identities. It would be good to see analyses that support the cutoffs for assignment above species level, i.e. 80% within genus. Refer to supplementary material of Ershova et al. (2021) (https://doi.org/10.1093/icesjms/fsab223) to include barcode gap analyses to justify the match between percent identity and taxonomic level within each phylum. The COInr-Med db had over 320K sequences, so a barcode gap analysis at the phylum level should be informative.

Adding an additional assignment software that provides a probabilistic assignment score for comparison with the VTAM software may be preferable. The core results of the paper rely on the assignment method. This will support the main inference of the paper that the databases lead the change in assignment rather than the assignment method itself or a combination of the two.

Minor Comments:
Lines 144-145: It would be good to see a check from the authors to assign the less abundant ASVs to ensure that taxa were not missed due to this clustering step. It would be good to see how many ASVs were removed at this step, to see if it was enough to make a difference computationally and worth the potential for missed species.

Validity of the findings

Impact and novelty would be better stated as a comparison between a "global" and "local" databases and with addition of barcodes. The underlying data provided are robust but the validity of the taxonomic assignment method should be compared with another method.

The authors have not addressed the major implication of non-specific amplification of the chosen primer. It is well know the Leray primer produces extensive non-metazoan reads and unassigned sequences are partly due to this. Please address where applicable throughout the paper, and specifically here:
Lines 245-246: This is not necessarily all due to database coverage of species, rather, partly due to non-metazoan amplification of the Leray primer. See reference Collins et al. (2019) (https://doi.org/10.1111/2041-210X.13276) and address in discussion and introduction.

Lines 268 – 274: This last paragraph should be extensively re-worked. Firstly, as noted, the majority of unassigned reads are likely not metazoans that can be barcoded. Secondly, it is strongly recommended to not advocate for increased barcoding of individuals that will not fill in the gaps of reference database coverage to meet real metabarcoding reads. As mentioned previously, COI is only one gene of many that are used in biodiversity assessment using metabarcoding. It is limiting to the field to advocate for single gene gap-filling when the technology is very accessible to generate more useful mitogenomes for reference databases.

Additional comments

What is the reason for only barcoding the COI region, and not sequencing the whole mitogenome sequence by genome skimming, which is only slightly more expensive. Metabarcoding studies for general biodiversity normally employ more than one marker from more than one mitochondrial or nuclear ribosomal DNA gene region. The additional information from a complete mitogenome would assist future studies and metabarcoding efforts that target different genes and make the reference databases for eDNA even more complete. Barcoding was appropriate for it’s time but does not meet the needs for metabarcoding, where other genes are known to be more informative for certain taxa and do not carry the non-specific amplification problems of the mutationally saturated COI gene region.

---

## Round 0.2 · accepted · Accept

One reviewer was able to re-review the manuscript following major revisions, and the reviewer and I agree that the manuscript has been substantially improved, especially by the addition of the comparative assignment methods. I believe this manuscript can now be accepted for publication.

A couple of minor points - when submitting the final proofs of the manuscript, I suggest specifying which are the forward and reverse primers when listing the primer sequences. Also in Figure 1, it would be beneficial to have the abbreviations in the boxes described in the figure caption.

Otherwise, the manuscript is well written, and I commend the authors for a nicely revised manuscript.

Reviewer 2 ·

Basic reporting

The authors have addressed all comments and have added some comparative taxonomic assignment methods.

Experimental design

The authors added three assignment methods that enhanced the quality and scope of the paper.

Validity of the findings

The additional information and insights gained from comparing taxonomic assignment methods with differently refined databases is tremendous and I applaud the authors on a job well done. The results presented here will elevate the field more than compared to the first iteration of the manuscript with just a single assignment method.

The access and provenance of the ASV data set used for analyses have now been provided, which improves transparency and repeatability of the study.